# Medical Devices, Invisible Women, Harmful Consequences

**DOI:** 10.3390/ijerph192114524

**Published:** 2022-11-05

**Authors:** Susan P. Phillips, Katrina Gee, Laura Wells

**Affiliations:** 1Departments of Family Medicine and Public Health Sciences, Queen’s University, Kingston, ON K7L 5E9, Canada; 2Department of Biomedical and Molecular Sciences, Queen’s University, Kingston, ON K7L 5E9, Canada; 3Department of Chemical Engineering, Queen’s University, Kingston, ON K7L 5E9, Canada

**Keywords:** sex differences, gender, medical devices, surgical mesh, hip prosthesis, women’s health

## Abstract

In this commentary, we explore the disproportionate risk women experience with the insertion of various medical devices. Although pre-market device testing and complication tracking could be improved for all, a failure to consider sex differences in hormones, anatomy, inflammatory responses, and physical function puts women at particular risk. This invisibility of women is an example of gender bias in medical science and practice, a bias that could be corrected in the ways we suggest.

## 1. Background

A medical device is “a contrivance designed and manufactured for use in healthcare, and not solely medicinal or nutritional” [1]. Medical devices, and particularly those such as a stethoscope or otoscope, that remain outside the body, might be considered ‘tried and true’. However, there is growing awareness that although implanted devices have a greater potential to harm, they may not undergo consistent and obligatory testing for safety and effectiveness [2] prior to their regulation, marketing, and use [3,4]. A recent public outcry about the insufficient scrutiny of device safety preceding insertion in patients may lead to better practices and an increased protection for all patients. The failure to recognize women’s disproportionate harm from implanted devices, however, remains. We hypothesize that the source of this oversight is a ubiquitous social myth that has permeated medicine, that women are just smaller versions of men with a few different but unimportant parts. This commentary explores that myth and its link to medical device development, testing, use, and failures. To draw readers in, we note that a 2019 examination of 340,000 reports to the US Food and Drug Administration (FDA) of device-related injuries or deaths found that 67% of those who were harmed were women [3]. In itself this statistic is concerning, but more surprising is the complicity (perhaps unintended) of North American regulators in making women invisible. Only clever sleuthing by journalists uncovered the disproportionate risk women face, because when it comes to reports of harm from medical devices, the FDA considers sex to be confidential [5]. 

The invisible woman is not unique to medicine. It is a myth woven throughout society. Days before a scheduled all women spacewalk, female astronauts were taken off a NASA mission when it became apparent that only one woman’s space suit had been made. A few years ago, a Swedish community had an ‘aha’ moment about snow ploughing and recognized that by prioritizing clearing roads before sidewalks, they were essentially making women’s activities invisible because women tended to walk (and to walk their children to school) while men drove. When they reversed the order of snow clearing, injuries and emergency room visits decreased significantly [6]. 

The ‘stories’ of several devices, and particularly of surgical mesh, illustrate an overall failure to assess the safety and effectiveness, but also a lack of consideration as to how sex shapes anatomy, immunity, hormonal milieu, and the function that allows for a normal adult to be equated with men and renders women invisible.

Surgical mesh has been used to repair hernias for almost a century. The ancestry, approval, and use of the product for this purpose highlights how underlying problems with the relationship between industry and medicine, coupled with limited regulatory oversight, hid evidence of a disproportionate harm to women [7]. The FDA’s role is to ensure that devices function as intended, that is, that they correct the problem for which they are the intended treatment. Complications or side effects are not part of defining this success, although they do weigh into measures of safety [8]. Inserting surgical mesh to repair hernias predated 1976, when the FDA began regulating devices. Those devices in use prior to that date were ‘grandfathered in’, that is, they were automatically approved with no need to produce reports of pre-market or clinical testing. Since 1976, the FDA has allowed device manufacturers to avoid regulatory scrutiny if they can demonstrate the substantial equivalence of a new product with an existing ‘predicate’ device [9]. Devices are classed according to risk. Although level III devices (highest risk) now require some pre-market clinical trial evidence of their utility and safety, this condition can be waved to reduce industry cost or time needed to bring the device to market. [8] The vast majority of surgical meshes now in use were cleared via this 510(k) route, that is, as equivalent to predicates, generating a chain of descendent products based on one or two untested parent devices [9]. Recalls of, or advisories about a medical device, do not extend to its descendants. Because of all this, neither Health Canada (the Canadian regulatory body) nor the FDA, from which Canada generally takes its lead, have required pre-market evidence of the efficacy or safety of surgical mesh in men or women. There are, nevertheless, hundreds of post-operative observational studies of mesh used in hernia repairs that should offer some guidance as to the potential safety of their use in women to repair uterine prolapse or stress urinary incontinence. Most of these studies, however, exclude women entirely and, to the best of our knowledge, none disaggregate findings by sex, although globally women receive 20% of 20 million annual hernia repairs. 

It was only years after women began to report post-op complications from pelvic mesh (2006) that Health Canada asked surgeons and hospitals to voluntarily report pelvic mesh failures and complications. Given how many women described ongoing and disabling problems [5], the 94 reports received from 2006 to 2019 suggest that this monitoring was ineffective. In 2014, Health Canada issued a warning about pelvic mesh, and the Society of Obstetricians and Gynecologists of Canada (the professional organization for Canadian gynecologists) recommended against its use for posterior prolapse treatment. Finally, in 2019, Health Canada withdrew approval for the trans-vaginal implantation of mesh in posterior prolapse repair, four months after the FDA recommended against mesh for anterior repairs. To date, the FDA has not made the adverse events reports it has received about pelvic mesh public.

This is not a story of harm that happened to affect women, but rather of the neglect of sex differences that underpin risk. The post-op research of mesh use outside the pelvis for hernia repairs assumed that women were anatomically and biologically identical to men and therefore need not be studied. Recently, Health Canada did acknowledge the need to protect women from device harm when it established the Scientific Advisory Committee on Health Products for Women (2019). However, part of that committee’s 2020 statement regarding pelvic mesh downplays biologic sex differences and shifts the responsibility for complications to surgical technique and skill:

“… although it appears that meshes may be dangerous in the hands of those who are not qualified to use them, there may be patients who are benefiting from these meshes especially when implanted by highly skilled surgeons. This is a physician education issue more than a device issue [10].”

As with any surgical procedure, complications from mesh use are minimized when performed by experienced surgeons. However, women’s well-documented stronger immune responses to foreign materials [11] and the increased infection risk of trans-vaginal insertion must also be considered as explanatory factors. The underlying problem of minimizing the consequences of women’s greater inflammatory response to materials the body views as foreign remains. In summary, the marketing of hundreds of meshes was piggybacked onto a prior acceptance of a few untested products. These were then repurposed in the mid-1990 s for trans-vaginal pelvic insertion with no pre-use testing, no ongoing monitoring or mandatory reporting, no patient information about potential risks, no cross-jurisdiction communication of those risks, and with suggestions that pelvic mesh failures were often attributable to surgical inexperience rather than the products themselves [10]. Regulatory bodies in the US and Canada do not conduct independent tests and tend to rely on industry reports about safety as well as effectiveness [3]. In repurposing mesh for pelvic use, there was no systematic approach to gathering evidence about safety, complications, or deaths. All of this demonstrates a failure of oversight, and a weighing of costs and benefits that favors industry over patient safety. There was no required or facilitated communication about observed risks and solutions among players, and no sense of shared responsibility for outcomes. A failure to consider female biology rendered women invisible and characterizes the resulting gender bias.

Rising levels of cobalt ions, metal fragments, and pain following metal-on-metal hip replacements were reported decades before a 2011 advisory from the FDA to manufacturers to study post-market safety [12]. A higher than predicted wear led to shedding of metal fragments, pain, and an increased risk of cancers, all of which were more prevalent in women. Perhaps fearing lawsuits, four manufacturers issued voluntary recalls of their products. In 2012, Health Canada offered advice to physicians on how to manage patients with such hip replacements. Press releases did not, however, mention women’s greater risk. Either manufacturers and Health Canada did not think about women’s anatomy, biology, hormones and, in particular, gait, all of which predisposed to disproportionate failure and complications, or did not deem research evidence of disproportionate harm worth considering [13].

Although, sadly, women have achieved parity in mortality from heart disease, the same cannot be said for their representation in studies of cardiovascular devices. Despite decades of guidelines from funders that women must be equally represented in clinical trials and that findings should be disaggregated by sex, the FDAs 1994 Gender Bias Directive stated that pre-market approval data should differentiate safety and effectiveness by sex, although how this was to happen was unclear as the Directive did not mandate the inclusion of women in the trials. Among 123 studies (2000–2007) of 78 high risk cardiovascular devices, 28% did not report the enrollees’ sex [14]. Only 33% of the remaining 92 studies included women, and fewer than half of these sex-disaggregated outcome data. The pre-approval documentation for implantable cardioverter-defibrillators (ICDs) illustrates this gender blindness. Fifteen of 126 participants in the sole pre-market study were women [14]. The manufacturer’s erroneous dismissal of this disproportion as aligning with women’s lower relative risk of cardiovascular disease was accepted by the FDA, who then approved ICDs for general use. The manufacturer did promise a future trial on women, one that has never occurred. A subsequent, independent meta-analysis showed that among women, ICDs do not decrease all-cause mortality [14].

## 2. Sex and the Invisible Woman

As we have described above, beyond an approval and regulatory process that can favor commerce over safety, lies a more systemic problem: the invisible woman [6]. It was patients’ and journalists’ reports of thousands of women developing unrelenting and serious sequelae that finally prompted regulatory bodies to acknowledge the harms arising from surgical mesh implanted into women. Health Canada’s effort to address this by creating the Scientific Advisory Committee on Health Products for Women is well-meaning, while also highlighting the ‘othering’ of women. Would not good science and governance require the consideration of biologic and social variability when investigating the effectiveness and safety of any device? It is worth noting that there has never been a need for special investigations following the insertion of medical devices in men because men are the norm while women are other [15].

Women remain invisible and gender bias is insidious when requirements for safety and efficacy research that include women and men are lax, and when each device shortcoming or outright failure is considered as a unique and individual occurrence unrelated to possible group characteristics such as sex. Historically, medicine viewed women, if they were viewed at all, as men with a few added or missing parts, that is, as flawed men [6,16,17]. Although some attention has been paid to this, the problem persists. Women’s experiences are more likely to be dismissed as ‘complaints’, while men’s symptoms are taken more seriously [4]. A number of authors have documented the problems of a lack of both pre-market testing and clinical trials prior to the approval of medical devices, and of a lack of subsequent long-term monitoring [3,7,18,19]. Few have linked device complications and failures to the invisible woman. As long as the developers of devices, regulators, and physicians continue to act as though the human population is homogeneous, and to assume that reactions to foreign materials will not vary by sex, the gender bias of the invisible woman will persist.

## 3. A Safer and More Inclusive Approach: Making Women Visible

When device harm exceeds benefit, there is typically no single, catastrophic error. It is small additive breaches and omissions at each step on the path from device conceptualization to clinical outcome that produce colossal problems. Each player—whether the engineers developing a device, biologists who may test it in laboratory settings, regulators who approve its use, family doctors who refer patients to the surgeons who insert the device, and, of course, patients themselves—is isolated from the rest and takes responsibility for only their piece of a bigger picture. There is rarely ‘cross-talk’ among the players who may assume that failures stem from individual patient circumstances rather than from the product itself. Group characteristics (with groups based on, for example, sex, but also age, race, etc.) are rarely considered as sources of failure or complications. Initiative and structural change are needed to bring together all the players and enable them to work together to conduct rigorous, methodical, informed, and inclusive research. The cost, both monetary and to patients, of prevention is lower than the cost of restitution.

At present, real-world studies of devices are occurring in patients who are unaware of a lack of pre-existing safety clearances or of the absence of evidence of effectiveness [18]. Devices are often inserted by well-meaning but equally naive surgeons. Current mandates for industry and physicians to report problems are unworkable. How, for example, would a manufacturer access data about individual patient outcomes after mesh implantation? Similarly, hospitals would have no knowledge of harms arising months after such surgery. Unless family physicians receive and read operative notes, they will be unaware that pelvic mesh has been inserted and will more likely attribute post-operative complications to surgical technique, an individual patient’s ‘bad luck’, or possibly to a perceived female proclivity to complain [4]. An obligatory reporting system that is easily accessed, links reports from all players, including patients and across jurisdictions, and is monitored by regulators, is essential.

Additionally, there is the question of responsibility. Engineers can ‘invent’ devices and health researchers can test them in female and male tissues and animals. They can also adapt materials to suit a variety of immune responses. However, these scientists need input from others who are more aware of specific patient biology, pathology, and lived reality to know what to avoid, modify, or address. Primary care providers continue to refer patients to specialists trusting that those patients will be offered interventions that are safe and evidence based. Surgeons and patients continue to trust that by the time a device is stocked by a hospital, it has been tested in vitro, in vivo, and in clinical trials, and received regulatory clearance for safety. It is unclear who is responsible for failures and complications and there is no or only minimal conversation and collaboration among those involved.

Public protection could be better served if the current default assumption of safety were inverted to assume risk of devices until proven otherwise. Proof should require independent pre-clinical in vitro and in vivo testing, perhaps funded by the device manufacturer but carried out by non-company scientists without any industry interference and with obligatory public reporting of all findings. There could also be a requirement for clinical trials prior to approval, each with diverse participation, measurement of all outcomes and not just of the device’s success in correcting the identified deficit, and follow-up tracking of immediate and long-term risks. A registry of all patients implanted with a particular device could be developed to provide a ‘denominator’ when assessing the frequency of complications and failures.

Internationally, others have flagged problems and suggested potential solutions [3,4,5,7,19]. However, none has linked these to the ubiquitous (in society, medicine and basic science) invisible woman. The myth that women are merely smaller versions of men, with hormones that confound research, allows half the population to be ignored. Related to this myth is an assumption that women complain without justification. “In dozens of interviews with ICIJ (International Consortium of Investigative Journalists) reporters and our partners, women said that when they complained of illnesses and maladies they believed were tied to an implant, they weren’t believed [4]”.

With an awareness of the disproportionate harm devices have caused in women [13] and the more frequent dismissal of female post-operative symptomatology, Health Canada established the Scientific Advisory Committee on Health Products for Women. This was an important first step that may eventually eliminate gender bias in medical device development and use.

The etiology of medical device complications and failures is complex and multifaceted. Solutions will challenge the entrenched, traditional social and economic norms that serve industry and individual disciplines more than diverse populations and patients. Industry and, to a lesser extent, medicine, have vested interests in maintaining current practices and allowing patients who suffer harm to believe that this arises from bad luck rather than systemic and preventable error and oversight. Although we have detailed only a few examples, there is a growing literature documenting that gender bias is ubiquitous in a device’s design and its use [5,13,20]. While, ultimately, such inequities will only be corrected via changes in social norms, scientific safeguards could immediately minimize harm. This requires disrupting the status quo of current design, testing, regulatory, and clinical practice, which in turn requires acknowledging the role of sex and gender in shaping effective device design and medical practice. Not testing for safety and success in women and men, and assuming that each device failure is a unique, individual occurrence unrelated to group characteristics, epitomizes gender bias. Internationally, many have called for pre-approval clinical trials of medical devices, and for long-term monitoring [3,19]. None, however, has linked device complications to the myth of the ‘invisible women’. When populations are viewed as homogeneous, and, for example, sex differences in immune responses to foreign materials are ignored, this oversight becomes gender bias that easily infiltrates clinical practice and perpetuates gender-biased research.

## 4. Conclusions

Despite a pervasive blind spot in social norms, women can be made visible in science [5,13]. Engineers are able to develop products that account for differential reactions and inflammatory responses if they are aware that this matters. Enrolling women in clinical trials is a first step in enabling findings to be reported by sex. Pre-market testing on male and female tissue and in men and women, post-market sex-disaggregated surveillance reporting, a registry of all devices implanted, the mandatory tracking of adverse events, and preventing ‘work-arounds’ by researchers, regulators, and industry would all help make women visible, as was recommended three decades ago by the US and Canadian research funding agencies.

Why, then, is this still a problem?

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
