# Peer review of "Medical Devices, Invisible Women, Harmful Consequences"

_ijerph, 2022, doi:10.3390/ijerph192114524_

Round 1

Reviewer 1 Report

This is a very important topic that deserves to be highlighted. Therefore, I want to see it presented in the best way.

The authors imply that sex and device safety have not been previously addressed. It has, though it begs the question about why these issues persist, which is highly relevant to this paper.

e.g. https://www.ncbi.nlm.nih.gov/pmc/articles/PMC3941917/ https://jamanetwork.com/journals/jama/article-abstract/1105094 https://www.sciencedirect.com/science/article/pii/B978012817728000084X https://www.ahajournals.org/doi/full/10.1161/JAHA.118.010869 https://www.liebertpub.com/doi/abs/10.1089/jwh.2017.6760 https://www.jacc.org/doi/abs/10.1016/j.jcin.2018.10.048 https://link.springer.com/chapter/10.1007/978-3-319-77932-4_7 https://jamanetwork.com/journals/jama/article-abstract/198032 https://academicworks.cuny.edu/gj_etds/440/

There has been quite a lot of focus on sex in clinical and pre-clinical trials of pharmaceuticals (e.g https://www.ncbi.nlm.nih.gov/pmc/articles/PMC6137686/ https://www.liebertpub.com/doi/abs/10.1089/jwh.2019.8084 ). I'd like to see a link between the approaches in these and those that are relevant to devices. Perhaps a table comparing them? Also, might be good to add some references about the many biological differences that are likely to affect devices - though this could sidetrack a good opinion piece (so up to the authors).

The authors correctly identify a number of issues. I'd like to see them categorized - e.g. effects due to culture, systematic effects, biological sex differences, reporting flaws. Similarly, I'd like to see an itemized list of specific, actionable recommendations (e.g. box 3 https://www.ncbi.nlm.nih.gov/pmc/articles/PMC7027556/ ). The conclusion is good.

There were some small wording issues. line 175, p4 should be them not then; line 28 pg 1 should be its not it's. Throughout, the authors use "blindness". More inclusive terms would be ignorance or obliviousness.

I agree wholeheartedly with the thesis the authors present. This is a VERY important and under-examined issue that deserves highlighting. With a little tweaking, this could be an even more powerful article and it deserves publication.

Author Response

responses are attached

Reviewer 2 Report

I find the text very interesting and well written. However, I miss a reflection that is closely related to the question asked by the authors at the end of the Commentary: Why do women's bodies continue to be made invisible in health care?

I think that in the Introduction or as a new heading, an analysis of gender bias in health care: aetiology, scope and consequences for women's health should be included. 

I congratulate the authors on their choice of topic.

Author Response

responses are attached
